# The Functional Role of Long Non-Coding RNA in Myogenesis and Skeletal Muscle Atrophy

**DOI:** 10.3390/cells11152291

**Published:** 2022-07-25

**Authors:** Keisuke Hitachi, Masahiko Honda, Kunihiro Tsuchida

**Affiliations:** 1Division for Therapies against Intractable Diseases, Institute for Comprehensive Medical Science (ICMS), Fujita Health University, Toyoake 470-1192, Japan; 2Department of Biochemistry, Kindai University Faculty of Medicine, Osaka-Sayama 589-8511, Japan; mhonda@med.kindai.ac.jp

**Keywords:** long non-coding RNA, Myoparr, skeletal muscle atrophy, cachexia, sarcopenia

## Abstract

Skeletal muscle is a pivotal organ in humans that maintains locomotion and homeostasis. Muscle atrophy caused by sarcopenia and cachexia, which results in reduced muscle mass and impaired skeletal muscle function, is a serious health condition that decreases life longevity in humans. Recent studies have revealed the molecular mechanisms by which long non-coding RNAs (lncRNAs) regulate skeletal muscle mass and function through transcriptional regulation, fiber-type switching, and skeletal muscle cell proliferation. In addition, lncRNAs function as natural inhibitors of microRNAs and induce muscle hypertrophy or atrophy. Intriguingly, muscle atrophy modifies the expression of thousands of lncRNAs. Therefore, although their exact functions have not yet been fully elucidated, various novel lncRNAs associated with muscle atrophy have been identified. Here, we comprehensively review recent knowledge on the regulatory roles of lncRNAs in skeletal muscle atrophy. In addition, we discuss the issues and possibilities of targeting lncRNAs as a treatment for skeletal muscle atrophy and muscle wasting disorders in humans.

## 1. Introduction

More than 400 skeletal muscles are present throughout the human body and account for 30–40% of the body weight in the human adult. The coordinated action of the skeletal muscles enables body movement, exercise, and postural maintenance. Additionally, maintaining adequate skeletal muscle mass is important for a healthy lifestyle to maintain body temperature, homeostasis, metabolism, blood pumping, and myokine secretion. The skeletal muscle is also a highly plastic organ. The skeletal muscle mass decreases with immobilization, malnutrition, and injury [1,2,3]. Diseases such as cancers, cardiovascular diseases, neurodegenerative diseases, chronic kidney disease (CKD), and chronic obstructive pulmonary disease (COPD) reduce skeletal muscle mass [4,5,6,7,8]. Cancer-induced cachexia is a complex metabolic disorder characterized by marked muscular wasting and is implicated in approximately 30% of all cancer-related deaths [9]. Aging also causes the loss of muscle mass and strength, even if one does not encounter such adversities and diseases [10]. The reduction in skeletal muscle mass due to aging is called sarcopenia and results in bedridden status, dysphagia, and dyspnea. Sarcopenia is present in 9.9–40.4% of community-dwelling older adults and impedes healthy life maintenance [11]. Moreover, in the recent COVID-19 pandemic, loss of muscle strength as a consequence of viral infection is becoming a major problem, regardless of the underlying diseases [12]. Global social isolation and physical inactivity due to the COVID-19 pandemic are also responsible for muscle weakness, even in non-virus-infected groups [13,14]. Skeletal muscle atrophy and weakness decrease the quality of life due to reduced activities of daily living and increased mortality from diseases [15]. On the other hand, resistance training is effective in alleviating muscle atrophy by improving both the quantity and quality of skeletal muscle. Recent systematic reviews and meta-analyses of cohort studies have indicated that muscular training reduces the risk of mortality, cardiovascular diseases, cancers, and diabetes [16]. However, performing adequate training safely without injury is difficult for sick and elderly individuals who have lost physical fitness. Therefore, elucidating the pathophysiology of skeletal muscle atrophy caused by complex factors is required to develop safe and effective treatments for muscle atrophy.

The balance between the synthesis and degradation of skeletal muscle components is the primary factor that determines the skeletal muscle mass [15]. The insulin-like growth factor-1 (IGF-1)/Akt/mammalian target of rapamycin (mTOR) pathway is a positive regulator of skeletal muscle mass. When IGF-1 binds to its receptor on the plasma membrane through a sequence of phosphorylation reaction cascades mediated by Akt and mTOR, it activates p70 ribosomal protein S6 kinase (p70S6K) and suppresses 4E-BP1, which are regulators of protein synthesis [17,18]. Consequently, activated translational initiation and elongation, and increased ribosome biogenesis contribute to skeletal muscle hypertrophy. Stimulated Akt inhibits the nuclear translocation of FoxO transcription factors and prevents the expression of muscle-specific E3 ubiquitin ligases, MuRF1 and Atrogin-1/MAFbx [19,20,21]. The β-adrenergic pathway also enhances muscle protein synthesis but is known to be associated with adverse cardiac-related events [22]. Protein synthesis normally predominates over protein degradation, whereas in pathogenic conditions, several signaling pathways are stimulated, leading to the degradation of muscle proteins. Myostatin, a cytokine belonging to the transforming growth factor β (TGF-β) superfamily, is a well-known inducer of skeletal muscle atrophy. An important negative regulation of skeletal muscle mass by myostatin was first demonstrated by McPherron et al. in 1997, using myostatin-deficient mice [23]. The mice showed a 2-fold increase in muscle mass due to increases in both muscle fiber size (hypertrophy) and myofiber number (hyperplasia). Increased muscle mass has also been observed in other animals, such as cattle, sheep, dogs, goats, pigs, rabbits, and fish, with naturally occurring or artificially mutated *myostatin* genes [24,25,26,27]. In Japan, muscle hypertrophy of sea bream with inactivation of myostatin protein by genome editing technology is commercially available [28]. Spontaneous mutations in the *myostatin* gene resulted in increased skeletal muscle mass in humans [29]. Intriguingly, activin, another member of the TGF-β superfamily, induces muscle atrophy [30]. Activin appears to be more potent than myostatin in inducing muscle atrophy in primates [31]. Both myostatin and activin signaling contribute to the predominant proteolysis of muscle proteins via the transcription factors Smad2/3 [32]. In addition, nuclear factor kappa B (NF-κB) and glucocorticoid signaling negatively regulate skeletal muscle mass [18,33]. Eventually, this atrophy-related signaling results in the activation of the ubiquitin-proteasome system [34]. Under several pathogenic conditions, hyperactivation of autophagy, another proteolytic system, contributes to the reduction of skeletal muscle mass [35].

Over the past decade, emerging evidence has demonstrated that long non-coding RNAs (lncRNAs), which are not translated into proteins, serve as novel regulators of diverse biological processes [36]. lncRNAs are more than 200 nucleotides (nt) in length and are expressed not only from non-coding genomic DNA, including cis-regulatory regions, introns, 5′- and 3′- untranslated regions, intergenic regions, and repetitive sequences, but are also transcribed from the coding genomic DNA in the antisense direction [37]. A curated database of human lncRNAs contains 268,848 lncRNA genes as of November 2019 [38]. Dysregulated lncRNA expression is associated with cancers, cardiovascular diseases, and neurodegenerative disorders [39,40]. lncRNAs have pleiotropic functions; transcriptional and translational regulation, chromatin modification, mRNA stability, RNA splicing, and nuclear body architecture [37,41]. The lncRNA sequences tend not to be conserved among species. In contrast to the unique functional domains found in proteins, the characteristic functional sequences of lncRNAs have not been fully determined [41]. The expression levels of lncRNAs are generally lower than those of mRNAs, and they are localized predominantly in the nucleus. Depending on their unique nucleic acid sequences, lncRNAs directly interact with proteins or genomic DNA and regulate the expression of downstream genes in the nucleus [42]. In the cytoplasm, lncRNAs function as decoys against microRNAs (miRNAs) [43], which post-transcriptionally fine-tune gene expression through mRNA degradation and translation inhibition [44]. Genome-wide studies using next-generation sequencing technology have identified thousands of novel lncRNAs in skeletal muscle. Additionally, lncRNAs have provided new insights into the regulation of skeletal muscle cell proliferation and differentiation. Recent studies have revealed the molecular functions of lncRNAs in the regulation of skeletal muscle mass. In this review, we summarize recent findings regarding lncRNAs involved in skeletal muscle atrophy caused by cancer cachexia, aging, neurological diseases, disuse, and fasting. The mechanism of muscle atrophy by lncRNA *Myoparr*, which we discovered, has been described in detail. We also discuss the possibilities and limitations of using lncRNAs as a treatment for human muscle atrophy. The molecular roles of lncRNAs described in this review in the regulation of skeletal muscle mass are summarized in Figure 1 and Table 1.

## 2. Myogenic Differentiation-Related lncRNAs

During embryogenesis, myogenic progenitor cells, myoblasts, proliferate until they reach an appropriate number, which is a major predictor of future skeletal muscle mass. Following mitotic arrest, myoblasts begin to differentiate and fuse to form multinucleated myotubes. Subsequently, myotubes further mature into myofibers, which form skeletal muscle tissues [45]. The formation of skeletal muscle, known as myogenesis, is regulated by a multistep process orchestrated by many transcription factors [46]. The basic helix-loop-helix (bHLH) transcription factor Myf5 initiates myogenesis [47] and cooperatively works with the bHLH transcription factor MyoD to determine myogenic cell fate [48]. Cell cycle of myoblasts is arrested concomitantly with myogenic differentiation. MyoD induces the expression of cyclin-dependent kinase inhibitors, p21 and p57, and p53 family members to arrest the cell cycle in myoblasts [49,50,51]. MyoD also activates the expression of the bHLH transcription factor, myogenin, leading to the entry of myoblasts into the myogenic differentiation program [52]. Formation of multinucleated myotubes by myoblast-myoblast fusion is mediated by the muscle-specific transmembrane protein myomaker [53]. Another bHLH transcription factor, MRF4, and MEF2 family transcription factors are also involved in both cell specification and differentiation [54,55]. Many lncRNAs have been identified in myoblasts and myotubes and have been shown to regulate myogenic differentiation [56]. Subsequently, the role of these lncRNAs in the regulation of skeletal muscle mass was investigated. In this section, we introduce the molecular functions of these lncRNAs, focusing on the regulation of myogenic differentiation and skeletal muscle mass.

### 2.1. Myoparr

By analyzing the transcriptionally activated region around the *myogenin* locus during C2C12 differentiation using an RNA polymerase II (Pol II) binding signal, we identified a novel lncRNA from the *myogenin* promoter region and named it *myogenin* promoter-associated myogenic regulatory antisense lncRNA, *Myoparr* [57]. *Myoparr* is expressed in a head-to-head fashion along with the *myogenin* gene during myogenic differentiation of human and mouse myoblasts. Mouse *Myoparr* is a single exon lncRNA and the lengths of *Myoparr* are 1172 nt and 1167 nt in C2C12 cells and C57BL/6J mice, respectively. This difference in length was due to the different lengths of repeat sequences in *Myoparr*. In humans, two types of *MYOPARR* exist: one is a 1977 nt lncRNA (isoform 1) consisting of a single exon, and the other is a 2245 nt lncRNA (isoform 2) consisting of three exons (Figure 2A). Human *MYOPARR* isoform 1 is expressed in a head-to-head fashion along with the *myogenin* gene, as in mice. In isoform 2, the first exon is expressed in reverse orientation from the last exon of *myogenin*, the 2nd exon of *MYOPARR* is in the 2nd intronic region of *myogenin*, and the 3rd exon is essentially in the same position as in isoform 1.

The timing of *Myoparr* and *myogenin* expression were mutually correlated in human and mouse myoblasts, suggesting that *Myoparr* participates in the regulation of *myogenin* expression [57]. Our group showed that *Myoparr* knockdown drastically decreased *myogenin* expression at the transcriptional level and inhibited the myogenic differentiation of C2C12 cells. Although several promoter-associated lncRNAs are involved in the DNA methylation status of neighboring genes [58], *Myoparr* knockdown did not affect the DNA methylation status of the *myogenin* promoter region. Instead, the reduction in *Myoparr* expression decreased Pol II recruitment, histone H3 lysine 4 trimethylation (H3K4me3), and histone H3 lysine 27 acetylation (H3K27ac) levels in the *myogenin* locus. Mechanistically, *Myoparr* directly binds to the DEAD-box protein Ddx17, a transcriptional co-activator, and promoted protein–protein interactions between Ddx17 and the histone acetyltransferase PCAF during C2C12 differentiation. The binding of PCAF to Ddx17 is essential for the high transcriptional activity of Ddx17 [59]. *Myoparr* is mainly localized to the chromatin fraction and binds to the *myogenin* promoter. Therefore, *Myoparr* activates *myogenin* expression by promoting histone acetylation at the *myogenin* promoter via the Ddx17-PCAF complex. In addition to *myogenin* expression, *Myoparr* regulates cell cycle withdrawal by activating the expression of *miR-133b*, *miR-206*, and *H19* lncRNA, which promotes myoblast cell cycle withdrawal [60,61,62]. Thus, *Myoparr* promotes both myogenic differentiation and myoblast cell cycle withdrawal by activating *myogenin*, *miR-133b*, *miR-206*, and *H19* during C2C12 differentiation. Although the molecular function of *Myoparr* in myogenic differentiation of mouse myoblasts is clear, the role of *MYOPARR* in human myogenesis remains to be elucidated.

The functions of lncRNAs can be altered depending on their binding partners [42]. We recently identified heterogeneous nuclear ribonucleoprotein K (hnRNPK) as a *Myoparr* binding protein [63]. Unlike Ddx17, hnRNPK negatively regulates the expression of *myogenin* at the transcriptional level. Deleting the hnRNPK-binding region of *Myoparr* enhances *myogenin* promoter activity. Interestingly, hnRNPK knockdown caused morphological abnormalities in the myotubes. Since hnRNPK knockdown increased the expression levels of MyoD protein and myogenin, hnRNPK might contribute to preventing premature differentiation of myoblasts by restricting *Myoparr* function.

Although the *myogenin* gene is essential for skeletal muscle development [64,65], its expression is suppressed after myogenesis by innervation. Conversely, in denervated skeletal muscles, reactivated *myogenin* expression triggers the expression of E3 ligases MuRF1 and Atrogin-1 [66]. Therefore, dysregulated *myogenin* expression results in skeletal muscle atrophy. During myogenic differentiation, *Myoparr* shares the same promoter region as *myogenin*. Moreover, MyoD and TGF-β signaling directly regulate *Myoparr* expression through the *myogenin* promoter region [57]. Therefore, we hypothesized that *Myoparr* expression is also activated by denervation in adult skeletal muscles. Sciatic nerve transection in mice resulted in a 10–20% reduction in tibialis anterior (TA) muscle mass 3–7 days after treatment. In this situation, we found increased *Myoparr* and *myogenin* expression. Since *Myoparr* is essential for *myogenin* expression during myogenic differentiation, we next examined whether *Myoparr* contributed to the activation of *myogenin* in denervated skeletal muscles. RNA interference (RNAi)-mediated knockdown of *Myoparr* decreased *myogenin* expression at both the mRNA and protein levels in denervated muscles (Figure 1, right panel). Moreover, *Myoparr* knockdown attenuated denervation-induced muscle atrophy (Table 1). Increased *Myoparr* expression was observed in denervated muscles but not in other muscle atrophy conditions caused by cancer cachexia, hindlimb suspension, cast immobilization, fasting, and dexamethasone administration [67]. Therefore, *Myoparr* serves as a specific inducer of muscle atrophy caused by denervation.

To determine whether alleviated muscle atrophy by *Myoparr* inhibition was caused only by the cell-autonomous effect of preventing *myogenin* reactivation, or whether other mechanisms are also responsible, we performed a comprehensive gene expression analysis in *Myoparr*-depleted TA muscles by RNA-sequencing (RNA-Seq) analysis. *Myoparr* knockdown increased and decreased the expression of the 423 and 425 genes, respectively [68]. Among them, *Gdf5*, which encodes one of the bone morphogenetic protein (BMP) family members, was previously shown to alleviate muscle atrophy caused by denervation [69]; therefore, we focused on the expression changes of *Gdf5* by *Myoparr* knockdown. Three days after *Myoparr* knockdown, substantially increased Gdf5 expression was confirmed by Western blot analysis of TA muscles (Figure 1, right panel). Moreover, *Myoparr* knockdown activated BMP signaling, as indicated by the increased phosphorylation levels of Smad1/5/8. Thus, our findings indicated that skeletal muscle atrophy caused by denervation may be regulated by the *Myoparr*/myogenin/Gdf5 axis (Figure 1, right panel, and Figure 2B). Intriguingly, the set of genes whose expression was affected by *Myoparr* knockdown differed substantially between myogenic differentiation and denervation-induced muscle atrophy [68], suggesting that in skeletal muscle atrophy conditions *Myoparr* functions with a binding protein other than Ddx17 or hnRNPK, which were identified as *Myoparr*-binding proteins during myogenic differentiation [57,63].

### 2.2. Charme

In 2015, *lnc-405* was first identified in differentiating C2C12 cells by RNA-Seq analysis [70]. Later, this lncRNA was renamed the chromatin architect of muscle expression (*Charme*) [71]. *Charme* is specifically expressed in skeletal and cardiac muscles, localized primarily to the chromatin fraction in the nucleus, and has orthologous transcripts in humans. *Charme* expression was observed after day 1 of myogenesis, and knockdown of *Charme* inhibited the myogenic differentiation of C2C12 cells. *Charme* is associated not only with its own transcribed genomic region but also with *Igf2*, *Tnnt3*, and *Tnni2* loci. Pol II recruitment and H3K9ac modification in these regions were reduced by *Charme* knockdown. *Charme* knockdown also caused physical dissection of the *Igf2* locus from the genomic region in which *Charme* itself was transcribed. It is noteworthy that forced overexpression of *Charme* by plasmid DNA did not rescue the defect caused by *Charme* knockdown. Therefore, *Charme* can promote myogenic differentiation by bringing its own transcribed genomic region close to the *Igf2* locus. Intriguingly, the binding of PTBP1, a splicing regulator, and MATR3, an RNA/DNA binding protein, to intron 1 of *Charme* was required for the localization of *Charme* to the nucleus [72]. In the gastrocnemius muscles of *Charme* knockout mice, expression levels of *MCK*, myosin heavy chain (*MyHC*), *Tnnt3*, *Tnni2*, and *Igf2* were reduced, resulting in skeletal muscle atrophy in 4-week-old mice (Figure 1, left panel, and Table 1) [71]. These knockout mice also exhibited aberrant heart size, shape, and hypertension. *Charme* knockout mice were not lethal and there were no reproduction problems, but they had a lifespan of less than one year.

### 2.3. Neat1

In the nucleus, several lncRNAs can form liquid droplet-like features, which are referred to as architectural lncRNAs [73]. Nuclear-enriched abundant transcript 1 (*Neat1*) is an architectural lncRNA, and its expression is increased during C2C12 differentiation [74]. Gain- and loss-of-function experiments have indicated that *Neat1* inhibits *p21* expression and promotes myoblast proliferation. At the same time, *Neat1* inhibited myogenic differentiation by suppressing *myogenin*, *Myh4*, *Tnni2*, and *Myomaker* expression. Unlike the well-recognized function of *Neat1* as an architectural lncRNA [75], *Neat1* interacts with enhancer of zeste homolog 2 (Ezh2), a component of the polycomb repressive complex 2, and recruited Ezh2 to the promoter regions of *myogenin* and *p21* in myoblasts [74]. During the lifetime of the mice, *Neat1* expression tended to increase until 4 weeks of age and then declined [76]. In adult mice, *Neat1* expression is increased in skeletal muscle during regeneration after cardiotoxin (CTX)-induced skeletal muscle injury [74]. Moreover, *Neat1* expression consistently increased in conditions of muscle atrophy, including denervation, hindlimb suspension, dexamethasone administration, and cast immobilization [67]. It is noteworthy that lentivirus-mediated knockdown of *Neat1* increased skeletal muscle mass in the TA, gastrocnemius, and quadriceps muscles with increased *myogenin*, *Myh4*, and *Tnni2* expression (Figure 1, left panel, and Table 1) [74]. However, *Neat1* inhibition delayed skeletal muscle regeneration following CTX injection in the gastrocnemius muscle, with reduced satellite cell numbers. Thus, inhibition of *Neat1* would be beneficial for inducing muscle hypertrophy, but detrimental to injured muscle.

### 2.4. TCONS-00036665

RNA immunoprecipitation using an Ezh2 antibody in the longissimus dorsi muscle of pigs revealed 356 Ezh2-binding lncRNAs [77]. *TCONS-00036665* was identified as an Ezh2-binding lncRNA in addition to *Neat1* and its expression increased during myogenic differentiation. *TCONS-00036665* is mainly localized in the nucleus. *TCONS-00036665* overexpression promoted proliferation but inhibited differentiation of pig skeletal muscle satellite cells. In contrast, *TCONS-00036665* knockdown inhibited the proliferation but promoted the myogenic differentiation of satellite cells, indicating that *TCONS-00036665* is required for satellite cell proliferation. Similar to *Neat1*, *TCONS-00036665* recruited Ezh2 to the promoter regions of *p21*, *myogenin*, and *Myh4*, thereby repressing the expression of these genes and maintaining the proliferative state of satellite cells. Interestingly, overexpression of *TCONS-00036665* in the skeletal muscles of the lower limbs of 6-week-old mice induced a decrease in muscle weight, accompanied by decreased expression of MyHC, MyoD, and myogenin proteins (Figure 1, left panel) [77]. Although the detailed molecular mechanism remains largely unknown, it is likely that *TCONS-00036665* negatively regulates postnatal muscle growth (Table 1).

### 2.5. linc-RAM

Conventionally, lncRNAs have not been considered to encode proteins, but several small polypeptides encoded by lncRNAs have been discovered [78]. The small polypeptides encoded by lncRNAs are called micropeptides. Myoregulin is a 46 amino acid micropeptide that is specifically expressed in the skeletal muscle [79]. Myoregulin directly interacts with sarcoplasmic/endoplasmic reticulum calcium Ca^2+^-ATPase (SERCA) and inhibits Ca^2+^ uptake into the sarcoplasmic reticulum. Mice lacking myoregulin show improved exercise performance. On the other hand, Yu et al. reported that an lncRNA encoding myoregulin functions not only to encode a micropeptide but also as a regulatory ncRNA [80]. They used MyoD chromatin immunoprecipitation (ChIP)-Seq data to identify 45 lncRNAs, whose expression was regulated by MyoD during C2C12 cell differentiation. One of them was named lincRNA activator of myogenesis (*linc-RAM*). Overexpression of *linc-RAM* promoted myogenic differentiation of C2C12 cells, while no effect was observed in *linc-RAM* mutants without the open reading frame of myoregulin. These findings suggest that the effect of *linc-RAM* in promoting myogenic differentiation is a feature of regulatory ncRNAs, independent of micropeptides. *linc-RAM* was localized in both the nuclei and cytoplasm of C2C12 myoblasts and myotubes. It has been hypothesized that the function of *linc-RAM* in the nucleus may be different from that of myoregulin. Using RNA immunoprecipitation and pulled-down assays, *linc-RAM* was found to be physically associated with MyoD [80]. Moreover, *linc-RAM* facilitated the association of the MyoD–Baf60c–Brg1 complex and enhanced the transcriptional activity of MyoD in the myogenic cells. *linc-RAM* knockout mice, which were generated by deleting exon 2 of *linc-RAM*, had fewer myofibers than wild-type mice (Figure 1, left panel, and Table 1) [80]. These mice also showed smaller myofiber size after 14 days of muscle regeneration following CTX injection. Notably, these mice retained intact myoregulin; therefore, reduced myofiber number and regeneration potential would depend on *linc-RAM* deficiency. The same group later found cytosolic *linc-RAM* function [81]. In myotubes, the majority of *linc-RAM* localizes in the cytoplasm, where *linc-RAM* interacts with glycogen phosphorylase (PYGM), which is involved in glycogen metabolism. Interestingly, PYGM promoted myogenic differentiation in an enzyme activity-dependent manner. *linc-RAM* positively regulated the enzymatic activity of PYGM by interacting with PYGM. Thus, crosstalk between lncRNAs and cellular metabolism may be a new regulator of myogenic differentiation. Other micropeptides have been identified in skeletal muscle cells and tissues [82]; therefore, it is expected that more examples of lncRNAs that function as both micropeptides and regulatory ncRNAs will be identified. Further studies are required to elucidate how the functions of both micropeptides and regulatory ncRNAs are differentially executed from one lncRNA.

### 2.6. lncMGPF

Murine lncRNA muscle growth-promoting factor, *lncMGPF*, is highly expressed in skeletal muscle tissues and promotes myogenic differentiation of C2C12 cells [83]. *lncMGPF* homologs have been identified in humans and pigs. *lncMGPF* knockout mice showed slower growth, smaller skeletal muscle, and weaker muscle regeneration than wild-type mice (Table 1). Overexpression of *lncMGPF* resulted in larger skeletal muscles than in wild-type mice, indicating that *lncMGPF* promotes skeletal muscle mass. Mechanistically, *lncMGPF* acted as a decoy for *miR-135-5p* and increased the expression of transcription factor Mef2c (Figure 1, left panel) [83]. Additionally, *lncMGPF* promoted human antigen R (HuR)-mediated mRNA stabilization of both *MyoD* and *myogenin* (Figure 1, left panel) [83]. Interestingly, 10 single nucleotide polymorphisms (SNPs) in pig *lncMGPF* have been identified in commercial and Chinese local pig breeds [84]. These SNPs are linked to *lncMGPF* stability and skeletal muscle growth. Taken together, *lncMGPF* may be a unique lncRNA with functional SNPs. However, further studies are required to determine whether *lncMGPF* antagonizes muscle atrophy.

## 3. lncRNAs Related to Muscle Atrophy and Hypertrophy Conditions

Sarcopenia is characterized by the age-related loss of muscle mass and skeletal muscle function [85]. A comparison of the expression profiles of lncRNAs in skeletal muscle biopsy samples from old and young participants identified 76 upregulated and 76 downregulated lncRNAs [86]. Using the 271 increased mRNAs and 153 decreased mRNAs discovered at the same analysis, Zheng et al. constructed a co-expression network between lncRNAs and mRNAs, and found that lncRNAs *AC004797*.1, *PRKG1-AS1*, and *GRPC5D-AS1* may be involved in aging-associated muscle atrophy [86]. Another group has shown that the expression of metastasis-associated lung adenocarcinoma transcript-1 (*Malat1*), which is a nuclear-retained lncRNA, decreased with age in mouse skeletal muscles [87]. Mikovic et al. also reported that *Malat1* expression decreases with age [76]. Additionally, aging decreased the expression of genome imprinting-related lncRNAs, *Gtl2*/*Meg3*, *Mirg*, and *Rtl1* [76]. Significantly decreased *Gtl2*/*Meg3* expression in skeletal muscle was also observed during skeletal muscle growth in 5–13-week-old mice [88]. Using four muscle atrophy models, cancer cachexia, denervation, cast immobilization, and fasting, it was found that the expression of 524, 283, 234, and 273 lncRNAs, respectively, was deregulated in each atrophy condition [89]. Of these, 51 lncRNAs showed common changes in expression across the four muscle atrophy models, suggesting that they might be associated with the onset or progression of muscle atrophy. Thus, the expression levels of lncRNAs have been revealed to be altered by muscle atrophy conditions or wasting disorders. In this section, we introduce the molecular functions of lncRNAs discovered under skeletal muscle atrophy or hypertrophy conditions.

### 3.1. Chronos

*Chronos*, previously referred to as *Gm17281*, is skeletal muscle- and heart-enriched lncRNA. Among striated muscles, *Chronos* is specifically expressed in fast-twitch myofibers [90]. *Chronos* expression largely decreased after CTX-induced muscle injury. In addition, a progressive increase in *Chronos* expression was observed with age, whereas *Chronos* expression was not altered by hindlimb suspension or streptozotocin-induced diabetic mice. The knockdown of *Chronos* resulted in myofiber hypertrophy, as shown by a 42% increase in myofiber cross-sectional area (CSA) in mice (Table 1). It is worth mentioning that the degree of this hypertrophy was comparable to that of siRNA-mediated myostatin knockdown [90]. *Chronos* knockdown increased *Bmp7* expression and led to the activation of BMP signaling, indicated by phosphorylated Smad1/5, which is an inducer of skeletal muscle hypertrophy in mice [69]. Inhibition of *Bmp7* expression counteracted muscle hypertrophy caused by *Chronos* inhibition. Mechanistically, *Chronos* suppressed *Bmp7* expression by recruiting Ezh2 to the upstream region of the *Bmp7* gene via the *Chronos* homology region [90]. Thus, *Chronos* negatively regulated skeletal muscle mass by suppressing *Bmp7* expression (Figure 1, right panel).

### 3.2. Atrolnc-1

Skeletal muscle loss due to cachexia increases the risk of morbidity and mortality in patients with cancer and CKD [4,7]. Sun et al. examined lncRNAs whose expression differed in three murine cachexia conditions, including cancer, CKD, and fasting, using microarray analysis. They identified 17 lncRNAs whose expression was dysregulated in the three cachexia conditions [91]. The expression levels of eight lncRNAs, *1110038B12Rik*, *Snhg8*, *Snhg1*, *Sngh4*, *Gm14005*, *Sox2ot*, *Bvht*, and *1700007L15Rik*, and nine lncRNAs, *A3300009N23Rik*, *1700020l14Rik*, *Airn*, *Meg3*/*Gtl2*, *Nctc1*, *Rian*, *H19*, *Neat1*, *Gt(ROSA)26Sor*, commonly increased or decreased in the three cachexia conditions, respectively. Among them, the expression of *1110038B12Rik* was robustly increased under these cachexia conditions; therefore, *1110038B12Rik* was named *Atrolnc-1*. Increased *Atrolnc-1* expression after fasting was confirmed and observed after dexamethasone treatment in mice [92,93]. Overexpression and knockdown of *Atrolnc-1* in C2C12 myotubes revealed that *Atrolnc-1* promoted proteolysis without affecting protein synthesis. Furthermore, overexpression of *Atrolnc-1* in TA muscles using an adeno-associated virus (AAV) vector induced muscle atrophy with increased MuRF1 expression in mice (Table 1). Knockdown of *Atrolnc-1* increased TA muscle weight in a CKD mouse model, whereas no effect on TA muscle weight was observed in sham control mice [91]. As a molecular mechanism, *Atrolnc-1* impeded ABIN-1, an inhibitor of NF-κB signal, and induced muscle atrophy by promoting NF-κB-mediated *MuRF1* transcription (Figure 1, right panel) [91]. Thus, inhibition of *Atrolnc-1* is a promising strategy for alleviating cachexia.

### 3.3. lncMAAT

The lncRNA muscle-atrophy-associated transcript *lncMAAT* is a promising therapeutic target for skeletal muscle atrophy. Li et al. first identified 1913 upregulated and 1117 downregulated lncRNAs in denervated gastrocnemius muscles compared with sham-operated muscles in mice using microarray analysis [94]. They further examined lncRNAs that play central roles in multiple types of skeletal muscle atrophy. They previously identified *miR-29b*, which is involved in multiple muscle atrophy [95,96], and then focused on lncRNAs that might interact with *miR-29b* through in silico analysis. This analysis identified 18 lncRNAs. Furthermore, four lncRNAs whose expression in skeletal muscle decreased after denervation were defined. Finally, *lncMAAT*, whose knockdown decreased C2C12 myotube diameter, was selected. The expression of *lncMAAT* was decreased in in vitro atrophy models by angiotensin II, H_2_O_2_, and TNF-α treatment in C2C12 myotubes. Additionally, *lncMAAT* expression was decreased in various types of muscle atrophy caused by angiotensin II infusion, fasting, immobilization, and aging in mice. No significant change in *lncMAAT* expression was observed in the heart atrophy model, indicating that decreased *lncMAAT* expression is specific to skeletal muscle atrophy. Interestingly, *lncMAAT* knockdown by two independent short hairpin RNAs (shRNAs) reduced the weight of the gastrocnemius muscles of mice (Table 1). In addition, overexpression of *lncMAAT* effectively alleviated muscle atrophy caused by angiotensin II infusion, denervation, and immobilization in mice. Overexpression and knockdown of *lncMAAT* decreased and increased *miR-29b* expression, respectively. Contrary to our initial hypothesis, *lncMAAT* did not directly bind to *miR-29b*. Instead, *lncMAAT* interacts with the transcription factor Sox6 and prevents Sox6 from binding to the *miR-29b* promoter, thus contributing to the repression of *miR-29b* expression at the transcriptional level [94]. *lncMAAT* is located in the third intron of *Mbnl1*, which encodes one of the pivotal splicing regulators. Li et al. showed that *lncMAAT* positively regulates the expression of its host gene *Mbnl1*. Since knockdown of *Mbnl1* reduced C2C12 myotube diameter, it is likely that *lncMAAT* regulates skeletal muscle mass by regulating *miR-29b* expression and RNA splicing events via *Mbnl1* (Figure 1, left panel) [94].

### 3.4. Pvt1

Alessio et al. examined differentially expressed lncRNAs using models for skeletal muscle atrophy caused by denervation and amyotrophic lateral sclerosis (ALS) by single-cell analysis [97]. The expression levels of lncRNA plasmacytoma variant translocation 1 (*Pvt1*), previously reported as a cell cycle regulator in carcinoma [98], were shown to increase under each muscle atrophy condition in mice [97]. *Pvt1* knockdown in proliferating C2C12 myoblasts upregulates the expression of mitochondria-related genes, resulting in a fragmented mitochondrial network. In vivo knockdown of *Pvt1* in the TA muscles of mice increased the size and number of mitochondria, as determined by electron microscopy [97]. Mitochondrial DNA content also increased after *Pvt1* knockdown. In denervated muscles, the amount of mitochondria was reduced, and *Pvt1* knockdown attenuated this reduction. Moreover, *Pvt1* knockdown increased the proportion of oxidative myofibers in gastrocnemius muscles. Intriguingly, the CSA of both fast and slow myofibers increased after *Pvt1* knockdown (Figure 1, right panel, and Table 1). Additionally, *Pvt1* knockdown prevented the denervation-induced atrophy of slow myofibers. Although the detailed molecular mechanism is unknown, *Pvt1* knockdown attenuated mitochondrial fragmentation, apoptosis, and autophagy via c-Myc phosphorylation and degradation [97].

### 3.5. LncEDCH1

RNA-Seq analysis of differentially expressed lncRNAs between a white recessive rock (a fast-growing broiler chicken) and Xinghua chicken (a slow-growing breed native to China) revealed that lncRNA *lncEDCH1* was enriched in hypertrophic broiler chickens [99]. *lncEDCH1* is evolutionarily conserved in wild turkeys and guineafowl. *lncEDCH1* was highly expressed in breast and leg muscles, whereas its expression gradually decreased during myogenic differentiation of chicken primary myoblasts. *lncEDCH1* has been shown to promote proliferation and inhibit differentiation of chicken primary myoblasts. Lentivirus-mediated *lncEDCH1* overexpression and knockdown in 1-day-old chickens revealed that *lncEDCH1* reduced intramuscular fat accumulation and inhibited muscle atrophy (Table 1). In addition, *lncEDCH1* knockdown increased the distribution of fast-twitch muscle fibers in chicken skeletal muscle. Mechanistically, *lncEDCH1* interacted with sarcoplasmic/endoplasmic reticulum calcium Ca^2+^-ATPase 2 (SERCA2) protein to increase the stability of SERCA2, which increases SERCA2 activity and enhances ER calcium accumulation (Figure 1, left panel) [99]. *lncEDCH1* also improved mitochondrial efficiency by activating the AMPK pathway. However, the mechanism by which *lncEDCH1*-induced reduction in cytosolic calcium leading to AMPK activation remains unknown. If the molecular function of *lncEDCH1* is evolutionarily conserved, *lncEDCH1* might be a potential target for the treatment of muscle atrophy and energy metabolism in humans.

## 4. Competing Endogenous lncRNAs Related to Skeletal Muscle Atrophy

In previous sections, the molecular mechanisms of lncRNAs in the regulation of muscle atrophy have been presented, mainly focusing on their interactions with specific proteins. However, lncRNAs also function by inhibiting the activity of miRNAs, that is, they act as competitive endogenous RNAs (ceRNAs) against miRNAs. For example, the first ceRNA reported in skeletal muscle cells was long non-coding MD1 (*linc-MD1*), which is expressed in the same genomic region as *miR-133b* and *miR-206*. The expression of *linc-MD1* increases during myogenic differentiation and is mainly localized in the cytoplasm [100]. The cytoplasmic accumulation of *linc-MD1* is favored by the HuR protein [101]. Moreover, the expression levels of *linc-MD1* are correlated with the progression of Duchenne muscular dystrophy [100,102]. Gain- and loss-of-function experiments showed that *linc-MD1* promotes myogenic differentiation. Cesana et al. found that *linc-MD1* inhibits *miR-133* and *miR-135* activities, and increases their target gene expression. Thus, *linc-MD1* is required for myogenic differentiation by acting as a ceRNA against *miR-133* and *miR-135* [100]. In addition to *linc-MD1*, *Meg3*/*Gtl2* regulates myogenic differentiation by acting as a ceRNA for *miR-135* [103]. Moreover, multiple ceRNAs for *miR-133* have been reported [104,105,106]. *lncMGPF*, mentioned above, is a ceRNA for *miR-135-5p* [83]. Considering these findings, many ceRNA-type lncRNAs have been discovered and shown to be associated with muscle atrophy. In this section, we introduce the molecular functions of lncRNAs acting as ceRNAs in the regulation of skeletal muscle mass.

### 4.1. H19

*H19* is an imprinted lncRNA expressed from a maternally inherited allele in mammals and is highly expressed in skeletal muscle tissues after birth. *H19* expression is induced during C2C12 cell differentiation. In vivo, H19 expression was increased or decreased by denervation or fasting in mice, respectively [67]. Several groups have reported the different molecular functions of *H19* during in vitro myogenic differentiation. Kallen et al. showed that *H19* contains binding sites for the *let-7* miRNA family and acts as a ceRNA for these miRNAs [107]. In addition, *H19* also worked as a source of *miR-675-3p* and *miR-675-5p*, which are encoded in the first exon of *H19* and regulate the expression of Cdc6 and SMAD family members (Smad1/5) during myogenic differentiation (Figure 1, right panel) [62]. However, whether *H19* promotes or inhibits muscle regeneration remains controversial [62,108]. In terms of the regulation of skeletal muscle mass, *H19* knockout mice exhibited muscle hypertrophy, accompanied by an increased number of muscle fibers (Table 1) [108]. Intriguingly, *H19* has recently been demonstrated to stabilize dystrophin protein by inhibiting ubiquitination induced by the E3 ubiquitin ligase Trim63 through direct binding to the dystrophin protein (Figure 1, right panel) [109]. Given that the expression level of *H19* is very high compared to that of other lncRNAs, the multifunctional role of *H19* in skeletal muscle function may be important. Further studies are needed to elucidate the molecular function of *H19* in regulating skeletal muscle mass.

### 4.2. lnc-mg

Zhu et al. identified a myogenesis-associated lncRNA (*lnc-mg*) that partially overlaps with the *Myh1* gene from the intergenic region between *Myh1* and *Myh4* in mice [110]. *lnc-mg* was highly expressed in skeletal muscle tissue and its expression increased during myogenic differentiation. Gain- and loss-of-function experiments revealed that *lnc-mg* promotes satellite cell differentiation. Interestingly, *lnc-mg* knockout mice had thinner muscle fibers and showed reduced muscle strength and endurance compared with wild-type mice [110]. Conversely, *lnc-mg* transgenic mice demonstrated thicker muscle fibers and greater muscle strength and endurance (Table 1). Moreover, myofiber diameters of *lnc-mg* transgenic mice were larger than those of wild-type mice even after denervation treatment. Mechanistically, *lnc-mg* functioned as a ceRNA against *miR-125b* to regulate the protein abundance of IGF-2 (Figure 1, left panel). Later, *lnc-mg* was shown to act as a molecular sponge for *miR-351-5p*, which inhibits C2C12 differentiation by targeting the *Lactb* gene (Figure 1, left panel) [111]. *lnc-mg* expression in the skeletal muscle was reduced after 48 h of fasting [93]. Thus, it would be interesting to investigate whether *lnc-mg* transgenic mice are also resistant to fasting-induced muscle atrophy.

### 4.3. SYISL

*SYISL*, a Synpo2 intron sense-overlapping lncRNA, was identified by microarray analysis as an lncRNA that is highly expressed in C2C12 myotubes [112]. *SYISL* overlaps with the fourth intron of the *Synpo2* gene, but unlike *Synpo2* expression, which is abundant in the stomach, small intestine, and heart, *SYISL* is highly expressed in muscle tissues such as the longissimus dorsi, leg muscle, and tongue. *SYISL* expression was directly activated by the MyoD protein via a MyoD-binding site (E-box) at its 5′ end. *SYISL* knockdown in differentiating C2C12 myoblasts increased the expression of genes associated with muscle differentiation, including *myogenin*, *Myh1*, *Myh2*, *Myh4*, *Myh7*, *Tnni1*, and *Mybpc2*, and decreased the expression of genes associated with axon guidance, cell cycle, and MAPK signaling pathways, including *CDKs*, *N-Ras*, *Zeb2*, *Igf2bp3*, *Ki67*, and *Pcna*. Further gain- and loss-of-function experiments of *SYISL* showed that *SYISL* promoted myoblast proliferation and myogenic fusion but inhibited myogenic differentiation. Mechanistically, *SYISL* recruits Ezh2 to the promoters of *p21*, *myogenin*, *MCK*, and *Myh4* to induce trimethylation at H3K27, leading to epigenetic silencing and regulation of downstream gene expression [112].

In *SYISL* knockout mice, the CSA of myofibers was reduced but the number of myofibers was increased, resulting in muscle weight gain at 2 months (Table 1) [112]. However, unlike at a young age, sarcopenia-induced muscle atrophy was alleviated in 18-month-old *SYISL* knockout mice compared with wild-type mice [113]. A similar alleviation of sarcopenia was also observed by shRNA-mediated suppression of *SYISL* in 18-month-old mice. In addition, *SYISL* knockout mice were resistant to dexamethasone-induced muscle atrophy [113] and these mice showed accelerated regeneration of skeletal muscle after CTX-induced muscle injury [112]. *SYISL* is conserved in humans and pigs, and overexpression of human and pig *SYISL* causes muscle atrophy in mice. The mechanism by which *SYISL* induces muscle atrophy is distinct from the regulation of muscle differentiation via Ezh2. *SYISL* acts as a sponge for *miR-23a-3p*, *miR-103-3p*, and *miR-205-5p*, leading to increased expression of the muscle atrophy-inducing genes *FoxO3a*, *MuRF1*, and *Atrogin-1* (Figure 1, right panel) [113]. Thus, SYISL is an lncRNA that is involved in muscle atrophy across species.

### 4.4. lnc-SEMT

Sheep enhanced muscularity transcript lncRNA (*lnc-SEMT*) was specifically expressed in skeletal muscle by RNA-Seq analysis in sheep [114]. Overexpression and knockdown experiments in sheep myoblasts revealed that *lnc-SEMT* accelerates myoblast differentiation. The authors generated transgenic sheep that specifically overexpressed *lnc-SEMT* in the muscles. These sheep exhibited a 1.5-fold increase in skeletal muscle weight (Table 1). In contrast, in vivo knockdown of *lnc-SEMT* reduced gastrocnemius muscle mass by approximately 2/3. *lnc-SEMT* increased IGF-2 expression by acting as a ceRNA for *miR-125b* shown by in vitro experiments (Figure 1, left panel). Moreover, increased IGF-2 was confirmed in the longissimus dorsi muscle of *lnc-SEMT*-transgenic sheep, indicating that the *lnc-SEMT*/*miR-125b*/IGF-2 axis increases skeletal muscle mass [114].

### 4.5. lncIRS1

Li et al. identified 239 lncRNAs, 763 mRNAs, and 101 miRNAs differentially expressed between hypertrophic and leaner broilers using RNA-Seq analyses [115]. They focused on the sponging function of lncRNAs against miRNAs and identified *lncIRS1*, which was highly expressed in the skeletal muscle of hypertrophic broilers, by constructing an lncRNA-miRNA-gene network. In particular, *lncIRS1* is primarily expressed in breast muscle and heart. During chicken embryogenesis, *lncIRS1* is expressed in somites, and *lncIRS1* regulates myoblast proliferation and differentiation of chicken myoblasts. Mechanistically, *lncIRS1* acted as a ceRNA for *miR-15a*, *miR-15b-5p*, and *miR-15c-5p*, and increased IRS1 expression (Figure 1, left panel). Moreover, overexpression of *lncIRS1* activated the IGF-1/Akt pathway by promoting Akt phosphorylation and increased the weight of chicken breast muscle by enlarging myofiber size (Table 1) [115]. Conversely, knockdown of *lncIRS1* induced muscle atrophy. Additionally, although the experiment was performed in vitro, overexpression of *lncIRS1* prevented muscle atrophy of myotubes induced by dexamethasone treatment. Taken together, the activation of *lncIRS1* shows therapeutic potential for the treatment of muscle atrophy.

### 4.6. lncMUMA

Mechanical unloading-induced muscle atrophy-related lncRNA (*lncMUMA*) was identified as the lncRNA that was most downregulated in murine skeletal muscle by hindlimb suspension [116]. Decreased *lncMUMA* expression is associated with reduced muscle mass after hindlimb suspension. Knockdown of *lncMUMA* reduced gastrocnemius muscle weight and strength. Intriguingly, inhibited myogenic differentiation with decreased *lncMUMA* and MyoD expression was observed in microgravity-conditioned C2C12 cells, which mimic in vivo hindlimb suspension. In C2C12 myotubes, *lncMUMA* prevented MyoD degradation by antagonizing *miR-762* activity as a ceRNA (Figure 1, left panel). MyoD expression was also decreased by hindlimb suspension. Therefore, it is hypothesized that *lncMUMA* maintains skeletal muscle mass by increasing MyoD expression as a ceRNA for *miR-762*. Zhang et al. generated *miR-762* knock-in mice that showed lower muscle mass and decreased MyoD expression in the gastrocnemius muscle. They demonstrated that overexpression of *lncMUMA* alleviated muscle atrophy in these mice by increasing MyoD expression [116]. Furthermore, *lncMUMA* overexpression reversed skeletal muscle mass, muscle function, and MyoD expression caused by hindlimb suspension in wild-type mice (Table 1). Thus, boosting the expression of *lncMUMA* would be beneficial for alleviating muscle atrophy caused by hindlimb suspension, and further application to other muscle atrophy models should be considered.

### 4.7. MAR1

Comparative analysis of expression changes of lncRNAs in skeletal muscle tissue between 6- and 24-month-old mice by microarray analysis revealed that lncRNA muscle anabolic regulator 1, *MAR1*, was downregulated in aged skeletal muscle [117]. In mouse organs, *MAR1* is highly expressed in the skeletal muscle, whereas modest *MAR1* expression is detected in the heart and kidney. In skeletal muscle, the expression of *MAR1* was decreased and increased by hindlimb suspension and fasting, respectively [93,117]. *MAR1* expression also increased during C2C12 differentiation. *MAR1* overexpression increased MyoD expression, whereas MAR1 knockdown decreased MyoD expression in C2C12 cells. *MAR1* promotes C2C12 differentiation by directly associating with *miR-487b*, which inhibits myogenic differentiation by targeting Wnt5a [117,118]. Intriguingly, overexpression and knockdown of *MAR1* increased and decreased skeletal muscle mass in mice, respectively (Table 1). Furthermore, *MAR1* overexpression in the murine gastrocnemius muscle alleviated muscle atrophy caused by sarcopenia and hindlimb suspension. Increased Wnt5a expression was observed in this model. Thus, *MAR1* may increase skeletal muscle mass by functioning as a ceRNA for *miR-487b* (Figure 1, left panel).

### 4.8. AK017368

*AK017368* is highly expressed in the skeletal muscle and heart, and to some extent in the lung. *AK017368* was localized in both the cytoplasm and nucleus of C2C12 cells, promoted proliferation, and inhibited the differentiation of C2C12 myoblasts by arresting them in the G0/G1 stage [119]. Knockdown of *AK017368* in gastrocnemius muscles increased *myogenin* and *MyHC* expression and induced muscle hypertrophy with increased myofiber CSA in mice (Table 1). Bioinformatic analysis revealed that *AK017368* has the recognition sequence of *miR-30c*, which is an inhibitor of proliferation and differentiation of C2C12 cells [120]. *AK017368* acted as a sponge for *miR-30c* in C2C12 cells (Figure 1, right panel). The expression level of Tnrc6a, one of the targets of *miR-30c* [120], was increased by *AK017368* overexpression. Therefore, the ceRNA function of *AK017368* against *miR-30c* may be required for myoblast proliferation. However, the detailed molecular mechanism by which *AK017368* inhibition causes skeletal muscle hypertrophy in mice remains unclear.

### 4.9. lnc-ORA

Cai et al. first identified the obesity-related lncRNA *lnc-ORA* as an adipogenic regulatory factor via regulation of the Akt/mTOR pathway [121]. Two years later, Cai et al. further demonstrated that *lnc-ORA* expression was upregulated with aging in mouse skeletal muscle tissues [122]. The expression levels of *lnc-ORA* also increased during the myogenic differentiation of C2C12 cells, and *lnc-ORA* was shown to promote proliferation and inhibit the differentiation of C2C12 cells. Interestingly, dexamethasone treatment increased *lnc-ORA* expression in C2C12 myotubes, and *lnc-ORA* knockdown restored dexamethasone-induced myotube atrophy, suggesting that *lnc-ORA* is a novel inducer of muscle atrophy (Table 1). Indeed, overexpression of *lnc-ORA* in murine skeletal muscles induced muscle atrophy with increased levels of atrophy-related proteins, MuRF1 and Atrogin-1, and decreased levels of muscle differentiation-related proteins, MyoD and MyHC. In both proliferating and differentiating C2C12 myoblasts, *lnc-ORA* was predominantly localized to the cytoplasm, where *lnc-ORA* acted as a ceRNA against *miR-532-3p* [122]. Phosphatidylinositol 3,4,5-trisphosphate 3-phosphatase and dual-specificity protein phosphatase (PTEN), an inhibitor of the Akt/mTOR pathway, was the target of *miR-532-3p*, thereby *lnc-ORA* overexpression increased the expression levels of PTEN protein in C2C12 myoblasts (Figure 1, right panel). Additionally, *lnc-ORA* interacts with IGFBP2, resulting in reduced stability of MyoD and MyHC proteins. Therefore, *lnc-ORA* is a novel modifier of myogenic differentiation and skeletal muscle mass through regulation of the Akt/mTOR pathway and myogenic transcription factors.

## 5. Skeletal Muscle Fiber-Type-Associated lncRNAs

Skeletal muscle is composed of a combination of slow- and fast-twitch muscle fibers, which have distinct metabolic and contractile properties [123]. Slow-twitch muscle fibers are rich in mitochondria and have high oxidative capacity, whereas fast-twitch muscle fibers have higher amounts of glycogen and produce ATP primarily through glycolysis. Skeletal muscle fibers exhibit remarkable plasticity in energy metabolism and contractile function to meet an individual’s activity and energy demands. Aging, inactivity, and wasting diseases reduce muscle mass and alter muscle fiber-type composition, along with changes in metabolic capacity. Skeletal muscle aging, represented by sarcopenia, primarily causes a decrease in the number and diameter of fast-twitch muscle fibers compared to slow-twitch fibers [10,124]. Additionally, the muscle wasting seen in cancer patients, similar to that seen in disuse atrophy, occurs primarily in slow-twitch muscle fibers, converts muscle fibers toward the fast-fiber type, and reduces muscle mass [125]. Thus, the regulation of muscle mass and alteration of muscle fiber-type composition can occur in parallel. Although enhanced catabolic signaling changes muscle fiber composition toward a slow-fiber type and prevents muscle hypertrophy, this does not fully explain the pathophysiology described above. Recent studies have identified lncRNAs as novel players involved in the regulation of muscle fiber-types. Therefore, a comprehensive understanding of the molecular mechanisms of lncRNAs underlying muscle fiber-type specialization and adaptation will provide therapeutic strategies for specific diseases. In this section, we introduce the functions and molecular mechanisms of lncRNAs involved in the regulation of muscle fiber-type switching and muscle mass.

### 5.1. Cytor

RNA-Seq analysis of differentially expressed lncRNAs in the human vastus lateralis muscle after one-leg knee extension exercise revealed cytoskeletal regulator RNA (*CYTOR*) [126]. *CYTOR* expression increases to some extent upon endurance exercise but is more pronounced in resistance training. *CYTOR* does not exist near (>100 kb separation) other genes and shows nucleotide conservation in mice (annotated as *Gm14005*) and rats (annotated as XR_146885.3). Increased expression of *Cytor* was observed in both mice and rats subjected to treadmill exercise, indicating that the *Cytor*’s response to exercise is also conserved among species. In human and mouse myoblasts, *Cytor* expression increased during myogenic differentiation. Gain- and loss-of-function experiments using C2C12 cells showed that *Cytor* inhibits myoblast proliferation and promotes myogenic differentiation. Young mouse gastrocnemius muscles with *Cytor* knockdown showed a sarcopenia-like phenotype, including muscle atrophy, decreased muscle strength, and decreased composition of fast-twitch fibers (Figure 1, left panel, and Table 1). On the other hand, restoring *Cytor* expression, which decreased with aging, recovered muscle weight loss, and increased muscle strength and fast-type fiber composition. Interestingly, overexpression of CYTOR in myoblasts derived from aged human muscles resulted in an improved myogenic differentiation potential and increased expression of fast-twitch myosin isoforms. Mechanistically, *Cytor*, by binding to the Tead1 transcription factor, reduced chromatin accessibility and occupancy in the binding motif of Tead and sequestered Tead1, thereby suppressing the slow-muscle phenotype and inducing a fast-muscle phenotype. Considering that Tead transcription factors and their cofactors are known to play an important role in the composition of muscle fibers [127,128,129], manipulating *CYTOR* function could have therapeutic potential to increase fast-twitch fibers and attenuate muscle atrophy caused by aging in humans.

### 5.2. lncRNA-FKBP1C

Analysis of differentially expressed lncRNAs between the breast muscle of white recessive rock and Xinghua chickens by RNA-Seq revealed *lncRNA-FKBP1C* [130]. During the differentiation of chicken primary myoblasts, *lncRNA-FKBP1C* expression transiently increased and then decreased. In vitro overexpression and knockdown experiments revealed that *lncRNA-FKBP1C* inhibits the growth of chicken primary myoblasts and promotes myogenic differentiation. Moreover, *lncRNA-FKBP1C* drove the slow-type muscle phenotype, both in vitro and in vivo. Moreover, *lncRNA-FKBP1C* overexpression induced an increase in muscle fiber diameter, whereas knockdown of *lncRNA-FKBP1C* reversed this phenotype (Table 1). Although the detailed molecular mechanism remains unknown, *lncRNA-FKBP1C* binds to Myh1b protein (homologous to murine embryonic MyHC) and enhances its stability (Figure 1, left panel) [130].

### 5.3. SMARCD3-OT1

In chickens, lncRNA *SMRCD3-OT1*, which is partly overlaid on the *Smarcd3x4* gene, was highly enriched in the breast and leg muscles [131]. *SMARCD3-OT1* expression remained constant in chicken primary myoblasts in a proliferative state but increased after the induction of myogenic differentiation. In chickens, *SMARCD3-OT1* expression increases with embryonic development and continues to be expressed in the skeletal muscle after birth. Overexpression and knockdown experiments in chicken primary myoblasts showed that *SMARCD3-OT1* promoted myoblast proliferation and myotube formation at the differentiation stage. *SMARCD3-OT1* also induces the expression of fast-twitch muscle fiber-related genes in myotubes. Moreover, *SMARCD3-OT1* induced hypertrophy and a fast-twitch muscle fiber phenotype in chicken skeletal muscles (Table 1). Thus, *SMARCD3-OT1* promotes cell proliferation and myotube formation, and can also induce fast-twitch muscle fiber-related genes in chicken primary myoblasts by increasing *Smarcd3x4* expression (Figure 1, left panel) [131]. *Smarcd3x4* is one of the isoforms of the evolutionarily conserved *Smarcd3* gene [131], which encodes a component of the SWI/SNF complex. Therefore, investigating whether *SMARCD3-OT1* and its molecular function are also evolutionarily conserved in humans is important.

### 5.4. ZFP36L2-AS

The ZFP36 ring finger protein-like 2 (ZFP36L2)-antisense transcript (*ZFP36L2-AS*) was more abundant in breast muscle than in leg muscle [132]. *ZFP36L2-AS* expression increased with chicken primary myoblast differentiation. *ZFP36L2-AS* suppressed proliferation and promoted myogenic differentiation of chicken primary myoblasts. *ZFP36L2-AS* also promoted glycolytic metabolism and suppressed oxidative metabolism by reducing mitochondrial function. However, these effects on cellular metabolism were not observed in adult satellite cells, suggesting the existence of a developmentally specific function for *ZFP36L2-AS*. Furthermore, *ZFP36L2-AS* knockdown in chicken skeletal muscle showed decreased expression of glycolytic metabolism-related genes, increased slow-twitch muscle fiber composition, and increased muscle mass with reduced expression of *MuRF1* and *Atrogin-1*, suggesting that *ZFP36L2-AS* induces a fast-twitch muscle fiber phenotype and muscle atrophy in vivo (Figure 1, right panel, and Table 1). *ZFP36L2-AS* bound to the acetyl-CoA carboxylase alpha (ACACA) protein and pyruvate carboxylase protein, and when *ZFP36L2-AS* was increased, the ACACA protein was activated with reduced phosphorylation levels, but pyruvate carboxylase was destabilized. Activated ACACA inhibits fatty acid β-oxidation and decreases pyruvate carboxylase, resulting in reduced mitochondrial function [132]. This may be one of the mechanisms underlying the induction of the fast-twitch muscle fiber phenotype by *ZFP36L2-AS* in chickens. *ZFP36L2-AS* is primarily conserved in birds [132]. Whether *ZFP36L2-AS* is a suitable therapeutic target for human muscle atrophy remains to be elucidated.

### 5.5. linc-MYH

The fast-twitch *Myh* genes are localized within a 300 kb region on chromosome 17 to form clusters in humans, and this genomic structure is conserved among species. Sakakibara et al. found a super-enhancer for fast-twitch *Myh* genes, 50 kb upstream of *Myh2* [133], and *linc-MYH* is located 4 kb downstream of the super-enhancer [133]. *linc-MYH* is specifically expressed in fast-twitch skeletal muscles and accumulates in the nuclei of adult mice. In vivo knockdown experiments using shRNAs showed that a decrease in *linc-MYH* expression was accompanied by a decrease in the expression of fast-twitch muscle fiber-related genes and an increase in the expression of slow-twitch muscle fiber-related genes. Conversely, forced expression of *linc-MYH* in slow-type muscle by in vivo transfection induced the fast-twitch gene *Myh4*. However, it was later reported that mice lacking *linc-MYH* showed a myofiber-type distribution similar to that of wild-type mice [134]. In addition, *linc-MYH* knockout mice had a larger pool of satellite cells than wild-type mice. The loss of *linc-MYH* may strengthen the association of chromatin remodeling proteins, including INO80, YY1, WDR5, and TFPT, leading to an increase in satellite cells. Moreover, in mice lacking the super-enhancer of fast-twitch *MyHC* genes, no clear change in myofiber-type distribution in distal hindlimb muscles was observed, whereas *linc-MYH* expression was lost [135], indicating that further studies are required to determine the function of *linc-MYH* in the regulation of myofiber-type distribution. Interestingly, *linc-MYH* knockout mice showed a muscle hypertrophy phenotype with increased muscle weight (Figure 1, right panel, and Table 1) [134]. We found *linc-MYH* expression was largely decreased in conditions of muscle atrophy induced by denervation, cast immobilization, fasting, and cancer cachexia [89]. Thus, decreased *linc-Myh* expression may be associated with the pathogenesis of muscle atrophy.

### 5.6. SMUL

Integrated transcriptomic and proteomic analyses identified 104 micropeptides translated from lncRNAs with altered expression during myogenic differentiation of chicken myoblasts [136]. SMAD-specific E3 ubiquitin-protein ligase (Smurf2) upstream lncRNA (*SMUL*) was one of the identified lncRNA-encoded micropeptides and was highly expressed in skeletal muscle tissues. *SMUL* expression was downregulated during chicken myoblast differentiation. Gain- and loss-of-function experiments showed that *SMUL* promotes myoblast proliferation and inhibits myogenic differentiation. In chicken skeletal muscle, *SMUL* induced muscle atrophy and activated switching from slow- to fast-twitch myofiber. Mechanistically, *SMUL* mediated the nonsense-mediated mRNA decay of Smurf2, which downregulates TGF-β signaling (Figure 1, right panel) [137]. Overexpression of Smurf2 induced muscle hypertrophy, whereas Smurf2 knockdown led to muscle atrophy (Table 1). Thus, *SMUL* reduced skeletal muscle mass by enhancing TGF-β signaling via Smurf2 stability. Given that myostatin and activin, members of the TGF-β superfamily, negatively regulate skeletal muscle mass and that Smurf2 is involved in their signaling [138], this mechanism can partly explain the effect of *SMUL*. However, it is unclear why the enhanced TGF-β signaling by *SMUL* stimulated the slow-to-fast fiber switch because depletion of myostatin, which is highly similar to TGF-β signaling, results in increased fast-twitch myofibers [139].

**Table 1 cells-11-02291-t001:** The Summary of lncRNAs involved in the regulation of skeletal muscle mass. Data are based on the in vivo experimental results, but partially include in vitro experiments for ceRNAs. ALS, amyotrophic lateral sclerosis; C.C., cancer cachexia; CKD, chronic kidney disease; Den, denervation; Dex, dexamethasone treatment; H.S., hindlimb suspension; S.M., skeletal muscle.

Name	Expression Changes by	Experiments in	Methods	For S.M. Mass	Function	Ref.
*AK017368*	-	Mouse	siRNA-mediated knockdown	Negative	Sponge for miRNA	[119]
*Atrolnc-1*	C.C., CKD, Dex, Fasting	Mouse	AAV-mediated overexpression shRNA-mediated knockdown	Negative	Transcriptional regulation	[91]
*Charme*	-	Mouse	Genetic knockout	Positive	Transcriptional regulation	[71]
*Chronos*	Aging	Mouse	siRNA-mediated knockdown	Negative	Transcriptional regulation	[90]
*Cytor*	Aging	Mouse	AAV-mediated overexpression, Gapmer-mediated knockdown	Positive	Transcriptional regulation	[126]
*H19*	Den, Fasting	Mouse	Genetic knockout	Negative	Source of miR-675-5p & miR-675-3p Dystrophin stability	[62,108,109]
*linc-MYH*	C.C., Den, Fasting, Immobilization	Mouse	Genetic knockout	Negative	Regulation of satellite cell pool	[134]
*linc-RAM*	-	Mouse	Genetic knockout	Positive	Micropeptide Transcriptional regulation	[79,80]
*LncEDCH1*	-	Chicken	Lentiviral-mediated overexpression Lentiviral-mediated knockdown	Positive	SERCA2 activity	[99]
*lncIRS1*	-	Chicken	Lentiviral-mediated overexpression Lentiviral-mediated knockdown	Positive	Sponge for miRNA	[115]
*lncMAAT*	Aging, Angiotensin II infusion, Den, Fasting, Immobilization	Mouse	Lentiviral-mediated overexpression Lentiviral-mediated knockdown	Positive	Transcriptional regulation	[94]
*lnc-mg*	Fasting	Mouse	Transgenic overexpression Genetic knockout	Positive	Sponge for miRNA	[110]
*lncMGPF*	-	Mouse	Lentiviral-mediated overexpression Genetic knockout	Positive	Sponge for miRNA mRNA stability	[83]
*lncMUMA*	H.S.	Mouse	Lentiviral-mediated overexpression	Positive	Sponge for miRNA	[116]
*lnc-ORA*	Aging	Mouse	AAV-mediated overexpression	Negative	Sponge for miRNA mRNA stability	[122]
*lncRNA-FKBP1C*	-	Chicken	Lentiviral-mediated overexpression Lentiviral-mediated knockdown	Positive	Protein stability	[130]
*lnc-SEMT*	-	Sheep	Transgenic overexpression shRNA-mediated knockdown	Positive	Sponge for miRNA	[114]
*MAR1*	Aging, Fasting, H.S.	Mouse	Transgenic overexpression shRNA-mediated knockdown	Positive	Sponge for miRNA	[117]
*Myoparr*	Den	Mouse	shRNA-mediated knockdown	Negative	Transcriptional regulation	[57,68]
*Neat1*	Den, Dex, H.S., Immobilization	Mouse	Lentiviral-mediated knockdown	Negative	Transcriptional regulation	[74]
*Pvt1*	Den, ALS	Mouse	Gapmer-mediated knockdown	Negative	Mitochondrial network regulation	[97]
*SMARCD3-OT1*	-	Chicken	Lentiviral-mediated overexpression ASO-mediated knockdown	Positive	Transcriptional regulation	[131]
*SMUL*	-	Chicken	Lentiviral-mediated overexpression Lentiviral-mediated knockdown	Negative	mRNA decay	[136]
*SYISL*	-	Mouse	Genetic knockout Lentiviral-mediated overexpression Lentiviral-mediated knockdown	Negative	Sponge for miRNA	[112,113]
*TCONS-00036665*	-	Mouse	Lentiviral-mediated knockdown	Negative	-	[77]
*ZFP36L2-AS*	-	Chicken	Lentiviral-mediated knockdown	Negative	-	[132]

## 6. Therapeutic Potential and Limitations of lncRNAs for Skeletal Muscle Atrophy in Humans

As described above, lncRNAs are potential therapeutic targets in human muscle atrophy and wasting disorders. However, several issues must be resolved before they can be applied to human therapy. First, even if the existence of corresponding lncRNAs among different species including humans is confirmed, the nucleotide sequences of lncRNAs are not generally well conserved compared to functional proteins [140]. Although they can alleviate muscle atrophy in mice and other vertebrates, whether their human counterparts have similar molecular functions in muscle mass remains unclear. Therefore, their applicability for treating muscle atrophy in humans must be carefully considered. The second issue is that optimal methods for controlling the expression or function of lncRNAs in human skeletal muscle have not been established. As mentioned above, increasing or decreasing the expression levels of lncRNAs is effective in regulating their functions. However, nucleic acid-based technologies, which have been applied to many mRNAs in the past, have shown limited progress in the treatment of human diseases [141]. Virus-vector-based technologies, which have been available for practical use in the last few years as COVID-19 vaccines [142], should be considered for lncRNA therapy in the future. In recent years, a small molecule-based inhibitor for *Xist* lncRNA has been developed [143]; therefore, the application of such a method may also be considered for other lncRNAs. The third issue is the complexity of the pathogenic factors of muscle atrophy [15], and lncRNAs, which function across the entire spectrum of muscle atrophy caused by cancer cachexia, disuse, fasting, and aging, have not yet been elucidated. Therefore, it is important to identify the cause of muscle atrophy and target appropriate lncRNAs.

Telomeres protect the ends of chromosomes from damage and a shortening of telomere length is known as a hallmark of cellular senescence. Telomere length is regulated by two lncRNAs, telomerase RNA component (*TERC*) and telomeric repeat-containing RNA (*TERRA*) [144]. Decreased *TERRA* expression in leukocytes is associated with sarcopenia [145]. Besides, *H19* impedes the function of telomerase, which extends the length of telomeres, in human acute promyelocytic leukemia cells [146]. Although whether the differences in telomere length are associated with sarcopenia is not conclusive [145,147,148,149], leukocyte telomere length is associated with a life span limit among humans [150]. In addition, genetic factors associated with longevity are being explored [151,152]; therefore, more detailed analysis of telomere-associated lncRNAs in skeletal muscles or research on lncRNAs associated with longevity could lead to identifying the new therapeutic targets for sarcopenia.

## 7. Conclusions and Future Perspectives

There are no lncRNAs in advanced clinical trial phases for skeletal muscle atrophy yet, but this may be because it has only been a decade since biological attention was focused on the pleiotropic functions of lncRNAs. Recent characterization has elucidated the association of lncRNAs with muscle atrophy. Thus, while many issues remain to be resolved, lncRNAs will become effective targets for skeletal muscle atrophy in the near future, leading to the development of therapeutic agents. We have high hopes that it will be possible to overcome human muscle atrophy using cutting-edge technologies related to new frontiers of lncRNAs.

## Figures and Tables

**Figure 1 cells-11-02291-f001:**
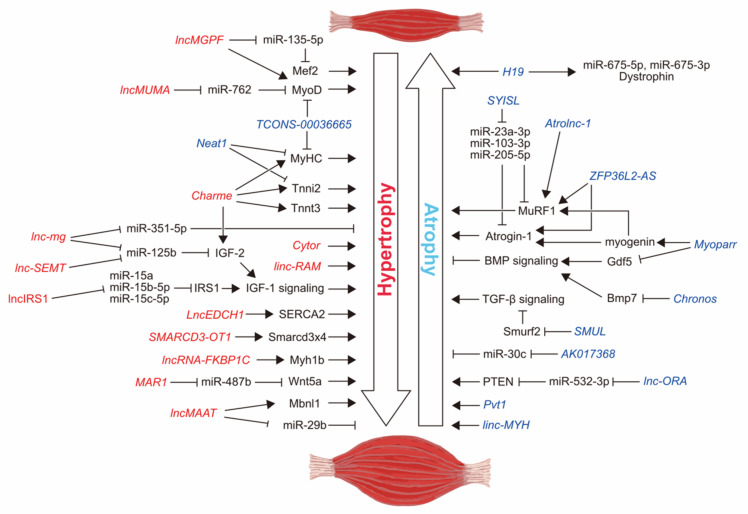
Schematic representation of the lncRNAs involved in the regulation of skeletal muscle mass. Based on the results from in vivo experiments (partial in vitro experiments are included for ceRNAs), lncRNAs increasing muscle mass are represented in red. lncRNAs decreasing muscle mass are represented in blue.

**Figure 2 cells-11-02291-f002:**
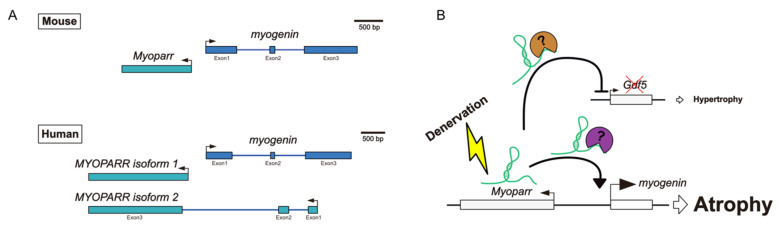
(**A**) Structures of both mouse *Myoparr* and human *MYOPARR*. (**B**) Molecular functions of mouse *Myoparr* in the regulation of muscle mass. Denervation activates *Myoparr* expression, and then *Myoparr* increases and decreases *myogenin* and *Gdf5* expression, respectively. Thus, *Myoparr* promotes muscle atrophy caused by denervation.

## Data Availability

Not applicable.

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
