# Peer review of "The Functional Role of Long Non-Coding RNA in Myogenesis and Skeletal Muscle Atrophy"

_cells, 2022, doi:10.3390/cells11152291_

Round 1
Reviewer 1 Report
Skeletal muscle plays an important role in animal movement, posture maintenance and metabolism. Muscle atrophy caused by sarcopenia and cachexia results in reduced muscle mass and impaired skeletal muscle function. The balance between the synthesis and degradation of skeletal muscle fibrin is the key to maintain skeletal muscle function. LncRNA play an important role in the process of muscle atrophy. In this study, the authors comprehensively review recent knowledge on the regulatory roles of lncRNAs in skeletal muscle atrophy. The following major concerns should be addressed before the article is accepted.
1. In the abstract, the description of “we comprehensively review recent knowledge on the regulatory roles of lncRNAs in skeletal muscle mass” should be “we comprehensively review recent knowledge on the regulatory roles of lncRNAs in skeletal muscle atrophy”.
2. In the third paragraph of the introduction, the author introduces that noncoding RNA (lncRNA, circRNA, miRNA) are involved in many important life processes, but the theme of this article is lncRNA. It is suggested to revise the description of the other two types of noncoding RNA, focusing on the research progress of lncRNA.
3. In the line 106 to 107, the authors described that “Unlike miRNAs and circRNAs, each lncRNA has pleiotropic functions”. It will be better to give several examples of lncRNAs which involvement in various biological functions.
4. The title of the manuscripts is “The functional role of long non-coding RNA in skeletal muscle atrophy”, but most of the content is related to myogenesis. Therefore, the title “The functional role of long non-coding RNA in myogenesis and skeletal muscle atrophy” should be more suitable.
5. LncMGPF plays important roles in the process of myogenic differentiation. However, no association of lncMGPF with muscle atrophy has been reported. It is suggested to move LncMGPF to the second part “Myogenic differentiation-related lncRNAs”.
6. The sixth part is relatively simple and needs further expansion.
Author Response
Skeletal muscle plays an important role in animal movement, posture maintenance and metabolism. Muscle atrophy caused by sarcopenia and cachexia results in reduced muscle mass and impaired skeletal muscle function. The balance between the synthesis and degradation of skeletal muscle fibrin is the key to maintain skeletal muscle function. LncRNA play an important role in the process of muscle atrophy. In this study, the authors comprehensively review recent knowledge on the regulatory roles of lncRNAs in skeletal muscle atrophy. The following major concerns should be addressed before the article is accepted.
[Our reply]
We appreciate the reviewer's favourable comments and kind suggestions. According to the valuable comments and suggestions, we revised the original manuscript and showed corrections highlighted using the Track Changes function in Microsoft Word.
Comment 1
- In the abstract, the description of “we comprehensively review recent knowledge on the regulatory roles of lncRNAs in skeletal muscle mass” should be “we comprehensively review recent knowledge on the regulatory roles of lncRNAs in skeletal muscle atrophy”.
[Our reply]
We appreciate your kind suggestion. We revised the sentence in the abstract.
Comment 2
- In the third paragraph of the introduction, the author introduces that noncoding RNA (lncRNA, circRNA, miRNA) are involved in many important life processes, but the theme of this article is lncRNA. It is suggested to revise the description of the other two types of noncoding RNA, focusing on the research progress of lncRNA.
[Our reply]
According to your suggestion, we revised the third paragraph of the introduction focusing on the research progress of lncRNA.
Comment 3
- In the line 106 to 107, the authors described that “Unlike miRNAs and circRNAs, each lncRNA has pleiotropic functions”. It will be better to give several examples of lncRNAs which involvement in various biological functions.
[Our reply]
Thank you for your suggestion. We added several examples of pleiotropic functions of lncRNAs (Please see Page 2, line 96-98).
Comment 4
- The title of the manuscripts is “The functional role of long non-coding RNA in skeletal muscle atrophy”, but most of the content is related to myogenesis. Therefore, the title “The functional role of long non-coding RNA in myogenesis and skeletal muscle atrophy” should be more suitable.
[Our reply]
According to the suggestion, we changed the title to “The functional role of long non-coding RNA in myogenesis and skeletal muscle atrophy” (Page 1).
Comment 5
- LncMGPF plays important roles in the process of myogenic differentiation. However, no association of lncMGPF with muscle atrophy has been reported. It is suggested to move LncMGPF to the second part “Myogenic differentiation-related lncRNAs”.
[Our reply]
According to the suggestion, we moved LncMGPF to the second part (Please see section 2-6., Page 9, lines 375-389).
Comment 6
- The sixth part is relatively simple and needs further expansion.
[Our reply]
According to the suggestion, we expanded the sixth part, and discussed about telomere and lncRNAs (Page 18, lines 871-882). Furthermore, we have added conclusions and future perspectives in the last section (Page 19, lines 883-891) according to the other reviewer’s comments. We hope the manuscript becomes easy to follow by expanding the sixth part and adding conslusions and future perspectives.
Reviewer 2 Report
In this Review Keisuke Hitachi and co-workers nicely described current knoledges on lncRNAs role in skeletal muscle 2 atrophy and sarcopenia. The Review is very well presented and sections are properly described. The initial graphical representation and the Tabelle help the reader throughout the manuscript.
This Reviewer has just some minor concerns that authors could address to ameliorate their Review:
1. It would be good to include a Final paragraphs with some Conclusions and future perspectives from authors according to the literature and their own experience.
2. It would be interesting to consider that lncRNAs, as epigenetic regulators, can influence sarcopenia also through the regulation of telomerase lenght. It would be interesting to include this as well, especially considering the numerous laboratory investigations coming now concerning the "immortalization of the DNA" related to the longevity of certain populations in the World. This could also be included in the last paragraph suggested at point 1.
3. If possible, it would be good to include a figure with those lncRNAs already used, or in advanced trial phases, to treat athrophy, including their mechanism of action and administration
Author Response
In this Review Keisuke Hitachi and co-workers nicely described current knowledges on lncRNAs role in skeletal muscle atrophy and sarcopenia. The Review is very well presented and sections are properly described. The initial graphical representation and the Table help the reader throughout the manuscript.
This Reviewer has just some minor concerns that authors could address to ameliorate their Review:
[Our reply]
We appreciate the reviewer's comments and kind suggestions. We also appreciate the comment stating "The Review is very well presented and sections are properly described". According to your valuable comments and suggestions, we revised the original manuscript and showed corrections highlighted using the Track Changes function in Microsoft Word.
Comment 1
- It would be good to include a Final paragraphs with some Conclusions and future perspectives from authors according to the literature and their own experience.
[Our reply]
We appreciate the reviewer's comment. We have added Conclusions and future perspectives in the last section (Page 19, lines 883-891).
Comment 2
- It would be interesting to consider that lncRNAs, as epigenetic regulators, can influence sarcopenia also through the regulation of telomerase length. It would be interesting to include this as well, especially considering the numerous laboratory investigations coming now concerning the "immortalization of the DNA" related to the longevity of certain populations in the World. This could also be included in the last paragraph suggested at point 1.
[Our reply]
According to your suggestions, we discussed about telomere length and lncRNAs, and genetic factors related to longevity and lncRNAs (Page 18, line 871-882).
Comment 3
- If possible, it would be good to include a figure with those lncRNAs already used, or in advanced trial phases, to treat atrophy, including their mechanism of action and administration
[Our reply]
We appreciate the reviewer's comment. As far as we know, lncRNAs are not yet at the clinical testing stage for muscle atrophy treatment. Therefore, we added the description of the clinical stage of lncRNAs for muscle atrophy in the Conclusions and future perspectives (Page 19, lines 884-886). We hope that lncRNAs will be targets of muscle atrophy in near future.